# The Selective CO_2_ Adsorption and Photothermal Conversion Study of an Azo-Based Cobalt-MOF Material

**DOI:** 10.3390/molecules27206873

**Published:** 2022-10-13

**Authors:** Li-Long Dang, De-Xi Zong, Xiao-Yan Lu, Ting-Ting Zhang, Tian Chen, Jiu-Long Sun, Jiu-Zhou Zhao, Meng-Yang Liu, Shui-Ren Liu

**Affiliations:** 1Henan Key Laboratory of Function-Oriented Porous Materials, College of Chemistry and Chemical Engineering, Luoyang Normal University, Luoyang 471934, China; 2Hubei Nuclear Industry Geological Bureau, Xiaogan 432003, China; 3School of Materials Science and Engineering, Zhengzhou University, Zhengzhou 450001, China

**Keywords:** metal–organic frameworks, porous material, crystal structure, CO_2_ adsorption, photothermal conversion

## Abstract

A new metal–organic framework (MOF), [Co_2_(L)_2_(azpy)]_n_ (compound **1**, H_2_L = 5-(pyridin-4-ylmethoxy)-isophthalic acid, azpy = 4,4′-azopyridine), was synthesized by a solvothermal method and further characterized by elemental analysis, IR spectra, thermogravimetric analysis, single-crystal and powder X-ray diffraction. The X-ray single-crystal diffraction analysis for compound **1** indicated that two *cis* L2^2−^ ligands connected to two cobalt atoms resulted in a macrocycle structure. Through a series of adsorption tests, we found that compound **1** exhibited a high capacity of CO_2_, and the adsorption capacity could reach 30.04 cm^3^/g. More interestingly, under 273 K conditions, the adsorption of CO_2_ was 41.33 cm^3^/g. In addition, when the Co-MOF was irradiated by a 730 nm laser, rapid temperature increases for compound **1** were observed (temperature variation in 169 s: 26.6 °C), showing an obvious photothermal conversion performance. The photothermal conversion efficiency reached 20.3%, which might be due to the fact that the parallel arrangement of azo units inhibited non-radiative transition and promoted photothermal conversion. The study provides an efficient strategy for designing MOFs for the adsorption of CO_2_ and with good photothermal conversion performance.

## 1. Introduction

The emission of CO_2_ will cause a rise in the global average temperature, which will lead to a series of ecological and environmental problems [1,2,3,4]. At present, there are two feasible methods for reducing CO_2_ concentrations. One is to develop new sources of energy that can be recycled, and the other is to use CO_2_ as a starting material to convert it into useful chemical products. However, disadvantages such as high cost and high energy consumption are the challenges in solving the above problems. Researchers have been trying to explore new recyclable adsorption materials for the adsorption and separation of CO_2_. Metal–organic frameworks (MOFs) as functional materials, due to their ease of design, diverse structure, chemical modifiability, high porosity and specific surface area, have been widely investigated in various fields such as fluorescent sensing [5,6,7,8,9,10,11,12], heterogeneous catalysis [13,14,15,16,17,18], drug delivery [19,20,21,22,23,24], gas storage and separation [25,26,27,28,29,30,31,32]. Modifying the metal sites and organic ligands is conducive to improving the adsorption and separation ability of the materials [33,34,35,36,37,38].

To date, numerous MOFs have been designed for the adsorption and separation of CO_2_ [39,40,41,42,43,44,45]. Luo et al. made many outstanding efforts in the field of gas adsorption [44,45,46,47,48,49]. For example, they designed a porous DMOF based on a photosensitive diarylethene ligand, and the photochromic behavior was controlled by UV irradiation with visible light for the first time [50], with up to a 75% CO_2_-desorption capacity. Additionally, they constructed another metal–organic framework material with a highly photosensitive structure by substituting diarylethene ligands for azo ligands, ECUT-15 [47]. Under UV irradiation, it still performed its general CO_2_ adsorption capability and had a highly selective adsorption of CO_2_ over N_2_, CH_4_, O_2_ and CO. At instantaneous conditions, the CO_2_ capture/release performance was up to 78%. In addition, M-MOF-74 series were reported by Matzger A. J. et al. and Long J. R. et al., respectively [51,52]. The results showed that M-MOF-74 had a highly selective adsorption capacity for CO_2_ under low pressure conditions due to the strong coordination between the unsaturated metal ions and CO_2_ in the structure There is an increasing number of MOFs supplied for small molecule gas adsorption and purification. However, the development of this research remains quite slow.

In addition, photothermal therapy has been demonstrated to have potential for cancer treatment because of the forceful killing of diseased lesions with high selectivity under near-infrared laser irradiation [53,54]. Except for purely inorganic materials or purely organic materials displaying the function of converting NIR light to heat, MOFs have also shown effective photothermal conversion capacity, which has been demonstrated by various ruthenium MOFs and Zn-MOFs based on π–π stacking interactions [55,56]. Although a variety of MOF complexes with near-infrared photothermal conversion properties have been achieved, reports about the photothermal effects induced by conjugation effects based on azo bonds are rare.

Herein, we report a new cobalt-based MOF, compound **1**, formulated as [Co_2_(L)_2_(azpy)]_n_ (H_2_L = 5-(pyridin-4-ylmethoxy)-isophthalic acid, azpy = 4,4′-azopyridine), which was synthesized under solvothermal conditions, and its crystal structure was determined by single-crystal X-ray diffraction (Figure 1). Compound **1** was characterized by the IR spectrum, elemental analysis, PXRD, solid-state UV–Vis absorption and thermogravimetric analysis. In addition, compound **1** exhibited a high selectivity for CO_2_ under 273 K conditions. The adsorption capacity could reach 41.33 cm^3^/g. In addition, obvious photothermal conversion performance was observed under 730 nm laser irradiation accompanied by a high conversion efficiency of 20.3% (temperature change: 53.6 °C).

## 2. Results

### 2.1. Crystal Structural Description

The single-crystal X-ray diffraction analysis indicated that compound **1** crystallized in a monoclinic *P*21/*c* space group. In the asymmetric unit (Figure 1), compound **1** contained two crystallographically independent Co(II) ions, namely Co1 and Co2 ions, and each Co^2+^ ion had a hexahedral coordination environment with four carboxylic oxygen atoms from three different H_2_L ligands, one pyridine nitrogen atom from the azpy ligand and one pyridine nitrogen atom from the L_2_^−^ ligand in the opposite direction. Careful analyzing of this structure showed that a methylene group serving as a flexible site was present in the H_2_L ligand, which caused the formation of a cage-type unit, in which the separation between two phenyl units was 6.92 Å. Furthermore, these cage-type units were bridged by multiple azpy ligands, finally resulting in the generation of a novel three-dimensional framework structure. Notably, a very large pore structure was observed in this 3D framework, showing a 6.3 Å × 11.7 Å window. Platon calculated a solvent-accessible void space of 2042.1 Å^3^ (equal to 40.6% of the cell volume), showing the potential porous MOFs of compound **1**. Moreover, the azpy ligands were arranged in parallel, in which the distance between two azo units was 3.42 Å, reflecting obvious π–π stacking interaction between the azpy ligands. In addition, an eight-membered ring was found based on two *cis* H_2_L ligands by the connection of two cobalt atoms (Figure 2). Additionally, the bond lengths were Co1 − O1 = 2.088(2) Å, Co1 − O2 = 2.300(19) Å, Co1 − O5 = 1.998(19) Å, Co1 − O7 = 2.035(18) Å, Co1 − N5 = 2.182(3) Å and Co1 − N6 = 2.148(3) Å. Furthermore, the adjacent non-bonding distance of Co1–Co2 was 4.35 Å.

### 2.2. PXRD and Thermal Properties

The simulated and experimental PXRD patterns of compound **1** are shown in Figure 3. Their peak positions corresponded well with each other, indicating the phase purity of the solids, although some small shifts were observed, which might be attributed to the cis–trans isomerism of the azo bonds [57,58].

In addition, the thermal stability of compound **1** was also investigated under a nitrogen atmosphere in the temperature range of 20–800 °C. From the TG curve (Figure 4), we found that the first weight loss occurred from 20 to 183 °C, which corresponded to the loss of the free water and DMF molecules. Additionally, there was almost no weight loss from 183 to 221 °C. Upon further heating above 221 °C, the successive losses corresponded to the collapse of the skeleton (352 °C) and the decomposition of the organic ligands (383 °C). In addition, the solid-state UV–Vis absorption of compound **1** was also explored carefully from 250 to 800 nm with a UV-vis spectrophotometer. The result clearly reflected a strong absorption from 300 to 550 nm. A careful structural analysis of this compound showed that the strong and wide-range UV-Vis absorption might be related to the absorption of azpy ligands due to the fact that the azo bond is a good chromophore, which can effectively absorb photoelectrons (Figure 4a) [9].

### 2.3. Adsorption Properties

The high chemical stability and thermal stability of compound **1** prompted us to study the gas adsorption performances. As shown in Figure 5a,b, the porosity of the activated materials of compound **1** was manifested by N_2_ adsorption at 77 K, expressing a fully reversible type-I isotherm, a signature characteristic of microporous materials, and giving a BET of 402.6 m^2^/g, a uniform pore size of 1.0 nm and a total pore volume of 0.21 cm^3^/g. Furthermore, this proper porosity and desirable aperture encouraged us to investigate the adsorption selectivity of CO_2_/CH_4_/CO in detail. At 298 K, the adsorption isotherms of the single components CO_2_, CH_4_ and CO were collected, as presented in Figure 6a. Compound **1** showed a unique adsorption capacity for CO_2_ compared to CH_4_ and CO. Furthermore, the adsorption amount of CO_2_ (30.04 cm^3^/g) was higher than that of CH_4_ (8.71 cm^3^/g) and CO (9.12 cm^3^/g). Finally, we further explored the adsorption capacity of compound **1** of CO_2_ under the influence of different temperatures (Figure 6b). The experimental results showed that the adsorption value could reach 41.33 cm^3^/g at 273 K. All the above experimental studies indicated that compound **1** could be used as a potential adsorbing material for CO_2_ in the future (Table 1).

### 2.4. Photothermal Conversion Properties

To our knowledge, π–π stacking interactions can trigger active non-radiative pathways and the inhibition of the radiative transition process, leading to an effective photothermal (PT) conversion phenomenon [59,60,61,62,63]. Furthermore, all the UV–vis spectra of compound **1** and two ligands displayed obvious near infrared absorption at 730 nm (Figure 4a). In addition, the TGA of compound **1** proved that compound **1** was able to maintain good structural stability until 221 °C, telling us that the temperature rise induced by the photo-thermal conversion was probably not the cause of the change in the structural framework (Figure 4b). Thus, the near-infrared photothermal conversion performance of compound **1** was performed earnestly under irradiation from a 730 nm near-infrared laser. In contrast, the temperature change experiments of the ligands H_2_L and azpy were also explored carefully under the same conditions as that of compound **1**.

Firstly, crystalline compound **1** was irradiated at a power of 0.6 W/cm^2^ under a 730 nm laser accompanied by the conditions of the external temperature being stable. Interestingly, a rapid temperature rise was observed. The temperature rose to 75.0 °C from 21.4 °C in only 109 s. Once the 730 nm near-infrared laser was turned off, the temperature declining process was seen clearly by the infrared imaging camera, reflecting the good photothermal conversion properties of compound **1** (Figure 7). Thus, the near-infrared photothermal conversion efficiency of compound **1** was calculated based on the reduced temperature data according to the calculation method described below (Equations (1)–(4)), showing a high conversion efficiency of 20.3%, reflecting the excellent near-infrared photothermal performance of Co−MOF, i.e., compound **1**. In contrast, illumination experiments of the ligands H_2_L and azpy under a 730 nm laser were also carried out. The results showed that the temperature increase in the ligands H_2_L and azpy was extremely weak and the temperature changes were 4.5 °C (from 22.1 to 26.6 °C) and 3.0 °C (from 22.4 to 25.4 °C), which was negligible (Figure 8). A careful analysis of structure **1** showed that the π–π stacking interaction between the azpy ligands could trigger non-radiative transition and induce obvious photothermal conversion effects.

Based on the total energy balance of the system:(1)$$\sum_{i}m_{i}c_{p,i}\frac{dT}{dt}=Q_{s}-Q_{\mathrm{loss}}$$
η = hS ΔT_max_/I (1 − 10^−A^) (2)
θ = (T − T_surr_)/(T_max_ − T_surr_) (3)

In Equations (1)–(3) above, *m_i_* (0.03 g) is the sample mass, c_p,i_ (1.926 J (g°C)^−1^) is the heat capacity of the system components and A is the absorbance of the samples at a wavelength of 730 nm (0.634). Q_s_ is the photothermal energy input by the near-infrared laser irradiation sample and Q_loss_ is the heat energy lost to the surrounding environment. I is the laser power (0.6 W cm^−2^). When the temperature reaches its maximum, the system is in equilibrium.
Q_s_ = Q_loss_ = hS ΔT_max_(4)
where h is the heat transfer coefficient and S is the surface area of the container. I is the laser power (0.6 W cm^−2^).

$\tau_{s}=\sum_{i}m_{i}c_{p,i}/\mathrm{hS}$, thus $\frac{d\theta }{dt}$ = $\frac{Q_{s}}{\tau_{s}\mathrm{hS}\Delta \mathrm{Tmax}}-\frac{\theta }{\tau_{s}}$, and when the laser is off, Q_s_ = 0, τ_s_ = −*t*/lnθ. The value of τ_s_ was obtained from the cooling curve of sample. The photothermal conversion efficiency η was 20.3%.

## 3. Experiment

### 3.1. Materials and Methods

All reagents and solvents were analytical grade and were obtained from commercial sources and used without further purification. Elemental analyses (C, H and N) were performed on a Vario EL III elemental analyzer. The infrared spectrum was measured with KBr pellets in the 4000–400 cm^−1^ region on a Nicolet 170SX spectrometer. Thermogravimetric analyses were performed on an SDT 2960 thermal analyzer from room temperature to 800 °C at a heating rate of 20 K min^−1^ under nitrogen flow. Powder X-ray diffraction (PXRD) data were collected on a Rigaku D/Max-2500PC diffractometer with Cu/Kα radiation (λ = 1.5406 Å) over a 2θ range of 5−50° with a scan speed of 5°/min at room temperature. CO_2_ (99.99%), N_2_ (99.99%), CH_4_ (99.99%) and CO (99.99%) gas adsorption measurements were performed using an ASAP 2020 system. UV–visible absorption spectra were recorded on a Lambda 750 s UV–vis spectrophotometer.

### 3.2. Synthesis of Compound ***1***

A mixture of H_2_L (27 mg, 0.10 mmol), azpy (18 mg, 0.10 mmol) and Co(NO_3_)_2_·6H_2_O (29 mg, 0.10 mmol) were dissolved in DMF (6 mL) and H_2_O (2 mL) in a screw-capped vial. The vial was capped and heated at 100 °C for 72 h. Orange block crystals of **1** were obtained in high yields (92%) based on the H_2_L ligand. Elemental analysis was calculated (%) for **1** C_38_H_26_Co_2_N_6_O_10_ (844.52): C, 54.04; H, 3.10; N, 9.95. It was found as: C, 54.07; H, 3.11; N, 9.50. The IR for **1** was: 3397 m, br, 3075 m, 2929 m, 2353 w, 1666 m, 1569 s, 1394 s, 1250 m, 1117 w, 1057 w, 920 vs, 845 w, 782 m, 727 m, 673 vs, 622 vs, 567 s, 528 vs, 479 vs.

### 3.3. Single-Crystal X-ray Structure Determination

The single-crystal X-ray diffraction analysis of **1** was carried out on a Rigaku Saturn 724 CCD diffractomer (Mo-Kα, λ = 0.71073 Å) at room temperature. The structure was solved by a direct method with SHELXS-97 [64] and refined by the full-matrix least-squares method on *F*^2^ with anisotropic displacement parameters for all non-H atoms (SHELXL-97) [65]. An empirical absorption correction was applied by the SADABS program [66]. The hydrogen atoms were assigned with common isotropic displacement factors and included in the final refinement with the use of geometrical restraints. The crystallographic data and selected bond lengths and angles for **1** are listed in Table 2.

Crystallographic data for the structural analysis were deposited at the Cambridge Crystallographic Data Center, and CCDC No. 2,178,932 for compound **1** contains the supplementary crystallographic data for this paper. These data can be obtained free of charge from the Cambridge Crystallographic Data Centre via www.ccdc.cam.ac.uk/data_request/cif (accessed on 10 September 2022).

## 4. Conclusions

To conclude, we indeed constructed a new cobalt-based MOF [Co_2_(L)_2_(azpy)]_n_ (H_2_L = 5-(pyridin-4-ylmethoxy)-isophthalic acid, azpy = 4,4′-azopyridine) by a solvothermal method. Notably, compound **1** had relative high CO_2_ adsorption capacities, namely 30.04 cm^3^/g and 41.33 cm^3^/g at 298 K and 273 K, respectively. However, the adsorption capacity of other gases was very weak (CH_4_: 8.71 cm^3^/g and CO: 9.12 cm^3^/g). Interestingly, compound **1** showed a rapid temperature change from 20.4 to 47.0 °C under a 730 nm laser radiation and the photothermal conversion efficiency could reach 20.6%, reflecting the good photothermal conversion properties of compound **1**. This study will promote the development of new photothermal conversion materials and gas adsorption materials.

## Data Availability

Not applicable.

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
