# Peer review of "The Selective CO_2_ Adsorption and Photothermal Conversion Study of an Azo-Based Cobalt-MOF Material"

_molecules, 2022, doi:10.3390/molecules27206873_

Round 1
Reviewer 1 Report (Previous Reviewer 1)
The paper can be accepted for publication after minor corrections:
1. Please change formula from [Co2(L)6(azpy)2]n to [Co2(L)2(azpy)]n in abstract. Similar correction needs to be done in table 1 and conclusions.
2. A word "cationic" needs to be removed from the last sentence of abstact, since the reported framework is uncharged. A word "selective" should be also removed from this line due to a moderate CO2 selectivity of the obtained material.
3. To replace 700 nm by 730 nm is possibly needed in the last sentence of introduction.
4. It is recommended to present PXRD pattern of MOF after laser irradiation to indicate a presence/an absence of structural changes during photothermal conversion process.
5. Crystallographic details should be marked as "Table 2", not Table 1.
Author Response
Please see the attachment.

Reviewer 2 Report (New Reviewer)
General comments:
In this manuscript, the authors report the synthesis of a new azo-based cobalt-MOF material. The MOF synthesized is unique and exciting. Also, this MOF exhibited a high adsorption capacity for CO2. This manuscript is not publishable in its present form due to the following reasons.
Comments on MS:
1. Compound 1 is obtained as a crystalline solid (Synthesis of Compound 1) and the crystal structure data in table 1 (which should be labeled as table 2) shows the presence of 3 H2O molecules {Table 1; Empirical formula C38H26Co2N6O10·3[H2O]}. However, the EA shows no signs of water molecules (Elemental analysis calcd (%) for 1 C38H26Co2N6O10 (844.52): C, 54.04; H, 3.10; N, 9.95. Found: C, 54.07; H, 3.11; N, 9.50). Explain this.
2. “From the TG curve (Figure 4), we found that the first weight loss of 11.1 % occurs from 20 to 183 °C, which corresponds to the loss of two free water molecules and one DMF solvent molecule (calcd: 11.5 %). which corresponds to the loss of two free water molecules and one DMF solvent molecule (calcd: 11.5 %)”. Table 1 (which should be labeled as table 2) shows only 3 H2O molecules and no DMF.
3. It is not mentioned in the manuscript that what form of compound 1 is used for measuring the properties i.e., crystalline or powder? Does the presence of 3 H2O molecules in the compound affect the adsorption properties?
4. PXRD and Thermal Properties “Although some small shifts can be observed, which might be attributed to the cis-trans isomerism of azo bonds” Cis-trans isomerism may cause structural changes in the MOF. Can authors confirm this speculation?
5. Provide the comparison with the known Co-MOFs in the literature Table 1.
6. Figure 4 and Table 1 are wrongly labeled (mentioned 2 times in the MS). Change the numbering of figures.
Corrections
1. The synthesized Co-MOF-1 is mentioned as compound 1, complex 1 and MOF 1 in the manuscript. Better to use one term for it.
2. The resolution of figure 1 is low.
3. Correct Co(ǁ) ions to Co(II) ions.
4. Correct igure 6 to Figure 6.
5. Check the MS for typos.

Round 2
Reviewer 2 Report (New Reviewer)
I think the revised paper is suitable for publication. However, before the final acceptance, minor revisions are required.
Comment on MS:
The crystal structure shows the presence of 3H2O molecules and the EA analysis was obtained with the dried sample. Moreover, the compound was dried before running the experiments. However, TGA analysis found 2H2O and a DMF molecule when the sample was not dried enough. How did the authors confirm this ratio of solvents (two free water molecules and one DMF solvent molecule) in TGA? Provide details in the text or remove the ratio of solvents from the text.
Author Response
Responses to editors:
We have made minor corrections in red color in our revised manuscript.
Responses to the Reviewers’ Comments
Reviewer: 2
Comments and Suggestions for Authors:
I think the revised paper is suitable for publication. However, before the final acceptance, minor revisions are required.
Response: We appreciate the reviewer’s positive comments very much.
- The crystal structure shows the presence of 3H2O molecules and the EA analysis was obtained with the dried sample. Moreover, the compound was dried before running the experiments. However, TGA analysis found 2H2O and a DMF molecule when the sample was not dried enough. How did the authors confirm this ratio of solvents (two free water molecules and one DMF solvent molecule) in TGA? Provide details in the text or remove the ratio of solvents from the text.
Response: Thank you very much for checking our manuscript carefully. The ratio of solvents from the text has been removed.

This manuscript is a resubmission of an earlier submission. The following is a list of the peer review reports and author responses from that submission.
Round 1
Reviewer 1 Report
Although the revised paper presents in a quite better shape, the novelty and significance of the reported material are still poor. There is only one ordinary MOF structure with questionable PXRD data, possessing absolutely ordinary (although authors claim otherwise at page 5 line 1) gas adsorption and selevtivity properties. No chemical stability was revealed in fact, although authors claim otherwise in section 2.3 line 1.
So, I can confidently repeat that the manuscript presents nothing interesting to be published in the high impact journal, such as Molecules. After correcting some pretentious statements (mentioned above in this review), I might recommend to sumbit it to some more crystal structure -oriented journal, for example Crystals or Molbank.
Author Response
Dear Reviewer:
Thank you very much for your precious advice, the manuscript has been revised accordingly.
Following are our answers and the list of changes, and we change this part to marked with yellow in the revised manuscript.
Reviewer 1: Although the revised paper presents in a quite better shape, the novelty and significance of the reported material are still poor. There is only one ordinary MOF structure with questionable PXRD data, possessing absolutely ordinary (although authors claim otherwise at page 5 line 1) gas adsorption and selevtivity properties. No chemical stability was revealed in fact, although authors claim otherwise in section 2.3 line 1.
So, I can confidently repeat that the manuscript presents nothing interesting to be published in the high impact journal, such as Molecules. After correcting some pretentious statements (mentioned above in this review), I might recommend to submit it to some more crystal structure-oriented journal, for example Crystals or Molbank.
Reply: Thank you for your suggestion. The manuscript has been rewritten carefully and some pretentious statements have been revised. The reviewer's suggestions are helpful in improving the quality of the manuscript. We will continue to explore carefully the adsorption performance of this MOF in future.

Reviewer 2 Report
Dear authors, I am sorry if the journal sent you revision tasks, but both reviewers rejected your manuscript in the first round, which means you should be notified to look for another journal.
Reviewer 3 Report
In this work, a new metal-organic framework was synthesized successfully by the solvothermal method. And the structure was characterized in detail by the single-crystal X-ray diffraction, IR, EA and TGA. From the structure analysis, the azo ligands connected these cyclic units in a parallel way by significant π-π accumulation, generating a 3D framework. In addition, the compound shows the meaningful selective CO2 sorption capacity for CO2 and CH4. This work is solid and well organized. Therefore, I recommend its publication in Molecules after minor revisions as below.
(1) There are some grammatical, spelling and reference format errors in the text, please check.
(2) There are multiple weight losses in the TG curve, therefore, the temperature point of each weight loss is important, please annotate them in Figure 4b.
(3) From the structure 1, a set of obvious parallel arrangements could be observed between two azo units. Do these units could promote selective CO2 sorption?
(4) Although the solid-state UV–Vis sorption of 1 was tested. However, the analysis for the data is inadequate. Please improve it carefully.
(5) In the section of 2.2, the authors said that although some small shifts can be observed, which might be attributed to the cis-trans isomerism of azo bonds. Appropriate references should be provided to demonstrate that azo bond could cause the shift of some PXRD peaks.
Author Response
Dear Reviewer:
Thank you very much for your precious advice, the manuscript has been revised accordingly. Following are our answers and the list of changes, and we change this part to marked with yellow in the revised manuscript.
In this work, a new metal-organic framework was synthesized successfully by the solvothermal method. And the structure was characterized in detail by the single-crystal X-ray diffraction, IR, EA and TGA. From the structure analysis, the azo ligands connected these cyclic units in a parallel way by significant π-π accumulation, generating a 3D framework. In addition, the compound shows the meaningful selective CO2 sorption capacity for CO2 and CH4. This work is solid and well organized. Therefore, I recommend its publication in Molecules after minor revisions as below.
- There are some grammatical, spelling and reference format errors in the text, please check.
Reply: Thank you for your valuable and thoughtful comments. The grammatical, spelling and reference format errors in the text have been checked and revised carefully.
- There are multiple weight losses in the TG curve, therefore, the temperature point of each weight loss is important, please annotate them in Figure 4b.
Reply: Many thanks for your advice. The temperature point of each weight loss has been annotated in Figure 4b.
- From the structure 1, a set of obvious parallel arrangements could be observed between two azo units. Do these units could promote selective CO2 sorption?
Reply: Many thanks for your advice. The good selective sorption property for CO2 molecules of complex 1 could be attributed to the stronger intermolecular interactions between CO2 molecules and the framework surface. In addition, the azo group has also a stronger affinity with CO2 molecules, thus promoting the selective CO2 adsorption performance.
- Although the solid-state UV–Vis sorption of 1 was tested. However, the analysis for the data is inadequate. Please improve it carefully.
Reply: Many thanks for your advice. The solid-state UV–Vis sorption of 1 has been reanalyzed.
- In the section of 2.2, the authors said that although some small shifts can be observed, which might be attributed to the cis-trans isomerism of azo bonds. Appropriate references should be provided to demonstrate that azo bond could cause the shift of some PXRD peaks.
Reply: Many thanks for your advice. The references have been provided to demonstrate that azo bond could cause the shift of some PXRD peaks. (Angew. Chem. Int. Ed. 2013, 52, 3695; Inorg. Chem. 2007, 46, 8490–8492)

Round 2
Reviewer 1 Report
I regret to state that no considerable improvement in the manuscript level is observed, so I cannot recommend the acceptance of this article.
As I mentioned in three earlier reviews, the presented material is not sufficient to be published in high-level journal, such as Molecules. Crystals or Molbank might be suitable journals for publication.